# Molecular and Photosynthetic Performance in the Yellow Leaf Mutant of *Torreya grandis* According to Transcriptome Sequencing, Chlorophyll *a* Fluorescence, and Modulated 820 nm Reflection

**DOI:** 10.3390/cells11030431

**Published:** 2022-01-27

**Authors:** Jianshuang Shen, Xueqin Li, Xiangtao Zhu, Zhicheng Ding, Xiaoling Huang, Xia Chen, Songheng Jin

**Affiliations:** 1Jiyang College, Zhejiang A&F University, Zhuji 311800, China; shenjianshuang18@163.com (J.S.); lxqin@zafu.edu.cn (X.L.); zxt8202@163.com (X.Z.); zuzims@163.com (Z.D.); JYXYhxl@aliyun.com (X.H.); comchenxia1212@yeah.net (X.C.); 2State Key Laboratory of Subtropical Silviculture, School of Forestry and Biotechnology, Zhejiang A&F University, Lin’an, Hangzhou 311300, China

**Keywords:** comparative transcriptome, photosynthetic electron transport, yellow leaf color mutant, gene expression, photosynthesis

## Abstract

To study the photosynthetic energy mechanism and electron transfer in yellow leaves, transcriptomics combined with physiological approaches was used to explore the mechanism of the yellow leaf mutant *Torreya grandis* ‘Merrillii’. The results showed that chlorophyll content, the maximal photochemical efficiency of PSII (*F_v_/F_m_*), and the parameters related to the OJ phase of fluorescence (φ_Eo_, φ_Ro_) were all decreased significantly in mutant-type *T. grandis* leaves. The efficiency needed for an electron to be transferred from the reduced carriers between the two photosystems to the end acceptors of the PSI (δ_Ro_) and the quantum yield of the energy dissipation (φ_Do_) were higher in the leaves of mutant-type *T. grandis* compared to those in wild-type leaves. Analysis of the prompt fluorescence kinetics and modulated 820 nm reflection showed that the electron transfer of PSII was decreased, and PSI activity was increased in yellow *T**. grandis* leaves. Transcriptome data showed that the unigenes involved in chlorophyll synthesis and the photosynthetic electron transport complex were downregulated in the leaves of mutant-type *T. grandis* compared to wild-type leaves, while there were no observable changes in carotenoid content and biosynthesis. These findings suggest that the downregulation of genes involved in chlorophyll synthesis leads to decreased chlorophyll content, resulting in both PSI activity and carotenoids having higher tolerance when acting as photo-protective mechanisms for coping with chlorophyll deficit and decrease in linear electron transport in PSII.

## 1. Introduction

*Torreya grandis* ‘Merrillii’, a variety of *Torreya grandis* Fortune ex. Lindt. (family *Taxaceae*), is a relict plant from the tertiary period and an important economic species that is widely grown in South China [1,2,3,4]. The seeds of *T. grandis* ‘Merrillii’ are processed to produce nuts that are served as a snack food with some health benefits [2,3,4]. A mutant of *T. grandis* ‘Merrillii’ with yellow leaves was recently discovered in a wild forest in Zhejiang province by our research team. The yellow leaf trait is stable through grafting and breeding. Plants with yellow leaves, particularly those that can be maintained year after year, play an important role in landscaping applications. The etiolation phenomenon commonly occurs in nature, and has been systematically studied in model plants and crops such as *Arabidopsis thaliana* [5], *Nicotiana tabacum* [6], and rice [7]. The molecular mechanisms of the yellow leaf color mutation have been summarized into several types, including gene mutations in the biosynthetic or degradation chlorophyll pathways, gene mutations that occur during chloroplast development and in functional pathways, and gene mutations that occur in the light signal transduction pathway [8,9,10]. Genes that are related to chloroplast development and function in mutants produce abnormal chlorophyll, resulting in the yellow leaf color phenotype. These yellow leaf color mutants are ideal material for an in-depth exploration of the development and function of chloroplasts, and provide excellent germplasm resources for genetic and breeding research [11]. Until now, the formation mechanism, photosynthetic energy, and electron transfer in the yellow leaves of woody plants have not been clearly elucidated.

Previous studies have reported that the composition, content, and distribution of pigments in mesophyll cells, chloroplast structure, and photosynthesis are directly related to leaf color in plants [12]. Biosynthesis of chlorophyll and photosynthesis occur in the chloroplasts of higher plants. Chlorophyll, combined with a specific protein, forms a photosynthetic complex that captures light and transfers electrons. Photosynthetic electron transfer occurs in the thylakoid membranes of chloroplast, and is related to photosystem I (PSI), photosystem II (PSII), and photosynthetic electron carriers [10,13,14,15]. Measurement of delayed chlorophyll *a* fluorescence (DF), prompt chlorophyll *a* fluorescence (PF), and modulated reflection at 820 nm (MR) is an important way to investigate changes in the photosynthetic electron transfer chain, including the PSI electron acceptor side, electron transport between PSII and PSI, and the PSII electron donor side [16,17,18,19]. To our knowledge, this method has not been used to study photosynthetic energy and electron transfer in yellow leaf mutant plants. Photosynthesis occurring in chloroplasts is an important issue in the field of biological research because it completes the conversion process of matter and energy and is the basis for living things. Therefore, research on the photosynthesis mechanism and its regulatory mechanism is needed.

High-throughput transcriptome sequencing (Illumina/Solexa) is widely used to analyze biological processes at the transcription level. High-throughput transcriptome sequencing technology lays the foundation for isolating and identifying the key genes of species without reference genomes, and can be used in molecular marker studies [20]. This technology has been used in several studies related to the molecular formation of yellow leaves in woody plants, including *Camellia sinensis* ‘Baijiguan’ [21] and *Lagerstroemia indica* [22].

The chlorophyll content and chlorophyll/carotenoid ratio, photosynthesis parameters, chlorophyll fluorescence, and expression of related genes were found to be different in green and yellow leaves, while the mechanism for formation of yellow leaves in different species is different [21,22,23,24]. Meanwhile, photosynthetic energy and electron transfer, and related gene expression in yellow leaf mutants of *T**. grandis* ‘Merrillii’ have not been studied. Therefore, we hypothesized that inhibiting chlorophyll synthesis could lead to decreased chlorophyll content and influence electron transfer in photosynthetic reaction centers (RCs), and that photo-protective mechanisms could help cope with chlorophyll deficit in the yellow leaves of *T**. grandis* ‘Merrillii’. This study was to research the pigments, chlorophyll *a* fluorescence, photosynthetic electron transfer, and photosynthetic and chlorophyll synthesis metabolic pathways in the yellow leaves of *T**. grandis* ‘Merrillii’, and explore the molecular and photosynthetic performance in order to provide a theoretical basis for the breeding, further research, and utilization of yellow leaves. Therefore, in this study, yellow leaf mutants of *T. grandis* ‘Merrillii’ were used as material to explore the physiological and molecular mechanisms of photosynthetic energy and electron transfer in its leaves by using physiological and transcriptome sequencing methods.

## 2. Materials and Methods

### 2.1. Plant Material

A total of 18 *Torreya grandis* ‘Merrillii’ grafting trees with green (*n* = 9) and yellow (*n* = 9) leaves were used in our study. In March 2017, these trees were established by using a 1-year-old *T. grandis* “Merrillii” scion on a 1-year-old *T. grandis* Fort. Ex Lindl. rootstock. Stem samples (scion) with green leaves and its natural mutant with yellow leaves (Figure 1) were collected from one normal plant and one mutant plant of *T. grandis* ‘Merrillii’. The rootstocks were cut about 5 cm above the root collar, and then grafted with scions. These trees were planted in individual pots (20 cm tall, 25 cm top diameter). Each pot was filled with 7 kg loam soil (field water holding capacity of 33%), watered when the pot was naturally dried (watered to saturation each time, about 0.5 L per pot), and fertilized per month with a half-strength Hoagland solution. Plants were placed outdoors at the Zhejiang A&F University (30°14′ N, 119°44′ E) in Lin’an, Zhejiang, China, and grown for three years under these conditions (annual average temperature is 15℃~23℃, annual solar emissivity is (419~502) × 10^4^ KJ/m^2^). On a sunny day on March 16, 2020, three replicates (three pots per replicate) of *T. grandis* ‘Merrillii’ with green leaves (WT) and three replicates of its natural mutant with yellow leaves (MT) were randomly selected and used as test materials in this study.

### 2.2. Measurement of Leaf Pigment Content

The mature third and fourth leaves from the top of the shoots were collected, weighed (0.1 g per sample), frozen in liquid nitrogen, and stored at −80 °C. The concentrations of chlorophyll *a* (Chl *a*), chlorophyll *b* (Chl *b*), and total carotenoids were determined using Lichtenthaler’s method [25,26]. WT and MT grafted *T. grandis* ‘Merrillii’ were randomly selected with three replicates, with three pots per replicate.

### 2.3. Simultaneous Measurement of PF, DF, and MR Kinetics

The measurements were made on the mature third and fourth leaves from the top of the shoots. The PF, DF, and MR kinetics were measured simultaneously after 30 min dark adaptation using a Multi-Function Plant Efficiency Analyzer (PE304NE, M-PEA, Hansatech, Norfolk, UK). A saturating light pulse (5000 µmol photons m^−2^ s^−1^) was emitted by the M-PEA. MR light reflection was measured separately with far-red light of 1000 µmol photons m^−2^ s^−1^. The emitter wavelengths were 635 ± 10 nm for the actinic light LED, 820 ± 25 nm for the modulated light LED, and 735 ± 15 nm for the far-red light LED. High-quality optical bandpass filters were used to protect the PF, DF (730 ± 15 nm), and MR (820 ± 20 nm) detectors. The PF, MR_820nm_, and DF were simultaneously measured when PF, DF, and MR light reflection and the light–dark conversion started 300 µs after exposure. The PF and MR light reflection signals were recorded in the light, and the DF signals were recorded in the dark [16]. The PF, DF, and MR kinetics curves were prepared in accordance with the methods described in previous studies [16,19,27]. Prompt chlorophyll fluorescent transients (OJIP transients) were quantified in accordance with the original data; minimal fluorescence intensity (F_o_, when all PSII RCs are open; recorded 20 µs after start of actinic illumination); maximum fluorescence intensity (F_m_, when all PSII reaction centers are closed); and fluorescence intensities at 300 µs (F_K_), 2 ms (F_J_), and 30 ms (F_I_). Some parameters were calculated from the measured OJIP transients [16,18,19], including the relative variable fluorescence at the J-step (V_J_), a trapped exciton moves an electron into the electron transport chain beyond Q_A_^−^ (ψ_o_), the efficiency required for an electron to be transferred from reduced carriers between the two photosystems to the PSI end acceptors (δ_Ro_), a given chlorophyll *a* molecule that functions as the PSII RC (γ_RC__)_, the quantum efficiency of electron transport at t = F_o_ (φ_Eo_), the quantum yield for reduction of the terminal electron acceptors on the PSI acceptor side (φ_Ro_,), the quantum efficiency of energy dissipation at t = F_o_ (φ_Do_), the Q_A_^−^ reducing active RCs per cross section (CS) at t = F_o_ (RC/CS_o_) and at t = F_m_ (RC/CS_M_), the absorption flux (ABS) of the antenna chlorophylls per RC (ABS/RC), the trapped energy flux per RC (TR_0_/RC), the dissipated energy flux per RC (DI_0_/RC), the electron transport flux per RC (ET_0_/RC), the electron flux-reducing end electron acceptors at the PSI acceptor side per RC (RE_0_/RC), the ABS per CS at t = F_m_ (ABS/CS_M_), the trapped energy flux per CS at t = F_m_ (TR_0_/CS_M_), the dissipated energy flux per CS at t = F_m_ (DI_0_/CS_M_), the electron flux per per CS (at t = F_m_) (ET_0_/CS_M_) and the electron transport flux per CS at t = F_m_ (RE_0_/CS_M_). DF is an obvious 20 s dark time point (the first reliable DF measurement value during each dark time interval), and it was selected during different time domains (from microseconds to minutes) to draw the DF-induced kinetic curve. The characteristic I_1_ was the first maximum of the DF curve (at 7 ms), I_2_ was the second maximum (at 100 ms), and D_2_ was the minimum of the curve. The MR kinetic was expressed as MR/MR_O_, where MR is the modulated reflected signal during illumination and MR_O_ is the first reliable MR measurement (taken at 0.7 ms). Three leaf replicates from wild-type and mutant-type *T. grandis* ‘Merrillii’ were randomly selected (three pots per replicate).

### 2.4. Transcriptome Sequencing

Three biological replicates of leaves from wild-type and mutant-type of *T. grandis* ‘Merrillii’ were randomly selected (three pot per replicate). Mature third and fourth leaves from the top of the shoots were collected, frozen in liquid nitrogen, and stored at −80 °C. Thus, there were six samples, with each sample containing the leaves (6–8 leaves randomly selected) from the three pots.

Total RNA extraction, RNA integrity evaluation and DNA libraries construction were made according to Han et al. [28]. The DNA libraries were sequenced using the Illumina HiSeq 2000 sequencing platform. RNA-seq, quality control, transcriptome assembly, gene functional annotation, gene expression analysis, and gene ontology (GO) and KEGG orthologue enrichment analyses were aided by the Novogene Biological Information Technology Co., Ltd. (Beijing, China) [28]. The thresholds for determining differentially expressed genes (DEGs) were set at |log2(Foldchange)| ≥ 1 and *p* value < 0.05. The differences in gene expression level were considered significant when a gene was identified as being differentially expressed. Multiple Benjamini–Hochberg tests had a false discovery rate (FDR) of 5% (*p* < 0.05) [29].

### 2.5. Quantitative Real-Time Polymerase Chain Reaction (qRT-PCR) Validation

Total cellular RNA was purified using Trizol RNA reagent (Life Technologies, Carlsbad, CA, USA) following the manufacturer’s instructions, and stored at −80 °C. qRT-PCR analysis was performed using the PrimeScript^TM^ RT reagent kit with gDNA Eraser (perfect real time; Code No. RR047; Takara, Shiga, Japan) following the manufacturer’s instructions. Both the qRT-PCR methods and cycling conditions of PCR reactions were used by Shen et al. [30]. Three technical replicates for each of the three biological repeats were performed for each sample using cyclophilin (*CYP*, Cluster-41647.20988) as the internal control. The gene-specific primers were designed using Primer 5.0, and are listed in Appendix A.

### 2.6. Statistical Analysis

One-way analysis of variance was carried out using Excel 2019 (Microsoft Inc., Redmond, WA, USA) and SPSS software (version 22.0; SPSS Inc., Chicago, IL, USA) for physiological data sets and to chart relevant parameters. Means were compared using the least significant difference test. *p*-values < 0.05 were considered significant.

## 3. Results

### 3.1. Analysis of Pigment Contents

As shown in Table 1, the Chl *a* and Chl *b* contents and the Chl *a*/*b* ratio in *T. grandis* ‘Merrillii’ leaves decreased significantly in the mutant-type plants. However, no differences in carotenoid content were observed between the mutant-type and wild-type plants.

### 3.2. Analysis of PF, DF, and MR Kinetics

The results in Figure 2A show that the yellow leaf mutants significantly changed the OJIP curve of the *T. grandis* ‘Merrillii’ leaves. The relative fluorescence intensities F_o_ of point O, F_J_ of point J, F_I_ of point I, and F_P_ of point P all decreased. In DF induction kinetics (Figure 2B), the fast phase occurs until 250 ms and includes the I_1_ (7 ms) and I_2_ (50 ms) peaks, which also coincide with PF’s OJIP transient stage. Meanwhile, the values of I_1_ and I_2_ decreased in the mutant-type leaves. The DF decay kinetics at I_1_ (Figure 2C) showed a significant decrease in the mutant-type leaves. The shape of the MR/MR_O_ kinetics in the mutant-type leaves of *T. grandis* ‘Merrillii’ demonstrated obvious changes (Figure 2D). The time of occurrence of the minimum value of the MR/MR_O_ kinetics in the mutant-type leaves appeared earlier than it did in the wild-type leaves. The lowest MR/MR_O_ kinetic points for the wild- and mutant-type leaves of *T. grandis* ‘Merrillii’ occurred in the range of 15–20 ms.

The PSI parameters were derived from MR/MR_O_ transients (Table 2), the maximum PSI oxidation rate (V_PS__I_) of the mutant-type leaves in the range of 0.7–20 ms decreased faster than that in the wild-type leaves. The maximum PS reduction rate (V_PS__Ⅱ__-PS__I_) of the mutant-type leaves in the range of 20–300 ms increased faster than that of the wild-type leaves. The PF parameters for the mutant-type *T. grandis* ‘Merrillii’ leaves were significantly influenced as shown in Table 3. In the mutant-type leaves of *T. grandis* ‘Merrillii’, the parameters F_v_/F_m_, ψ_o_, γ_RC_, φ_Eo_, φ_Ro_, and RC/CS_M_ all decreased significantly, the parameters δ_Ro_ and RC/CS_0_ did not show any significant change, while φ_Do_ increased significantly.

The absorbed energy fluxes are shown in Figure 3. The ABS/RC, TR_0_/RC, and DI_0_/RC value were all increased significantly in the leaves of the mutant-type *T. grandis* ‘Merrillii’, while ET_0_/RC and RE_0_/RC had no significant differences between the yellow and green leaves of *T. grandis* ‘Merrillii’ (Figure 3A). When all of these values were determined at the maximum (index M) chlorophyll fluorescence levels, the ABS/CS_M_ and TR_0_/CS_M_ achieved demonstrated no significant difference between the wild-type and mutant-type *T. grandis* ‘Merrillii’. The DI_0_/CS_M_ increased significantly in the leaves of mutant-type *T. grandis* ‘Merrillii’, while ET_0_/CS_M_ and RE_0_/CS_M_ were both significantly decreased in the leaves of mutant-type *T. grandis* ‘Merrillii’ (Figure 3B).

### 3.3. Overview of the Transcriptome Data by RNA-Seq Analysis

RNA-Seq data were generated from the leaves of wild-type and mutant-type *T. grandis* ‘Merrillii’, which were sequenced in three biological replicates. Excluding the adapters, low-quality regions (reads in which the base number of Q_phred_ < = 20 accounts for more than 50% of the whole read length), and all possible contamination, six paired-end libraries obtained more than 64.62 Gb of clean data, attaining an average of 143 million clean reads per library with Q30 > 94.06% and a GC percentage of 43.12–43.53% (Appendix A). All of the reads were deposited in the NCBI Sequence Read Archive under accession number PRJNA687396. As *T. grandis* ‘Merrillii’ does not have a reference genome sequence, Trinity [31] was used with the min_kmer_cov set to 2 by default and all other parameters set to the de novo assembly of all the clean reads as default, as shown in Appendix A. The Pearson’s correlation coefficient (R) between the biological replicates was >0.75 (Appendix A). More than 80% of the reads were mapped back to the assembled transcripts in the six samples (Appendix A) using RSEM software [32]. Nr, Nt, Pram, KOG, KO, Swiss-prot, and GO databases were used to annotate all unigenes, and a total of 89,955 unigenes were successfully annotated with gene information (Appendix A).

### 3.4. Pairwise Comparisons of the Transcriptomes between Wild Type and Mutant Type of T. grandis ‘Merrillii’

The comparisons of the unigenes annotated in the leaves of wild-type and mutant-type *T. grandis* ‘Merrillii’ are shown in Appendix A. A total of 39,986 (48%) unigenes were annotated in all the samples. To determine the gene expression response to the different leaf colors, differentially expressed genes (DEGs) in the pairwise comparison between the wild-type (WT) and mutant-type (MT) leaves were identified using the combined criteria log2 fold-change in the range between ≥ +1.0 and ≤ −1.0 and a corrected Padj value < 0.05. In total, we identified 10,789 DEGs in the comparison (Appendix A). In total, 9798 unigenes were upregulated and 991 unigenes were downregulated in the wild-type compared to the mutant-type leaves (Appendix A).

GO and KEGG pathway enrichment analyses were applied to explore the functions of the DEGs. The GO annotation suggested that the unigenes were annotated with 60 GO terms (Appendix A), in which more than 4000 unigenes were annotated in the ‘binding’, ‘membrane’, and ‘protein binding’ terms. The results indicated that these terms play crucial roles in the response of *T. grandis* ‘Merrillii’ to different leaf colors. A total of 117 pathways were categorized from the comparison in the KEGG pathway enrichment analysis. Among them, the top 20 enriched pathways (based on q values) in the comparison are shown in Appendix A. These pathways were closely related to ‘terpenoid backbone biosynthesis’, ‘ribosome biogenesis in eukaryotes’, ‘ubiquitin mediated proteolysis’, and ‘regulation of autophagy’. The significantly enriched terms had no direct relationship with leaf color in *T. grandis* ‘Merrillii’.

### 3.5. DEGs Involved in Photosynthesis and Photosynthetic Pigment Metabolism Associated Pathways

The metabolic pathways directly related to leaf color were analyzed, and these included the photosynthetic and the porphyrin and chlorophyll biosynthesis pathways. Three DEGs were clustered in the photosynthetic pathway based on the KO database from the comparison (MT vs. WT) and matched three genes, including the PSII P600 reaction center D1 protein (*PsbA*), PSII cytochrome b559 subunit alpha (*PsbE*), and ferredoxin-NADP^+^ reductase (*PetH*). *PsbA* and *PsbE* encode the D1 protein and β subunits of cytochrome b559 in PSII, respectively. *PetH* encodes the synthesis of the photosynthetic electron carrier ferredoxin-NADP^+^ reductase (*FNR*). The expression levels of the unigenes, including *PsbA* (Cluster-41647.24000) and *PsbE* (Cluster-41647.10995), in the yellow leaves of *T. grandis* ‘Merrillii’ were higher than those in green leaves, while the expression level of the *PetH* (Cluster-41647.15332) gene in the yellow leaves was lower than that in the green leaves.

Sixteen DEGs were clustered in the porphyrin and chlorophyll metabolic pathway based on the KO database developed from the comparison (MT vs. WT) and matched 12 genes, including hydroxyl methylbilane synthase (*hemC*), uroporphyrinogen decarboxylase (*hemE*), coproporphyrinogen III oxidase (*hemF*), protoporphyrin/coproporphyrin ferrochelatase (*hemH*), protoporphyrinogen/coproporphyrinogen III oxidase (*hemY*), cytochrome c oxidase assembly protein subunit 15 (*ctaA*), heme o synthase (*ctaB*), glutamyl-tRNA synthetase (*gltX*), cytochrome c heme-lyase (*HCCS*), glucuronosyltransferase (*UGT*), light-independent protochlorophyllide reductase subunit L (*chlL*), and geranylgeranyl diphosphate/geranylgeranyl bacteriochlorophyllide, a reductase. Among these genes, *hemC*, *hemE*, *hemF*, *hemY*, *gltX*, and *chlL* are involved in chlorophyll synthesis, and *hemH*, *HCCS*, and *hemH* are involved in cytochrome c synthesis. The expression levels of the genes *hemC* (Cluster-41647.4613), *hemE* (Cluster-29883.0), *hemF* (Cluster-17955.0), *hemY* (Cluster-31222.0), *gltX* (Cluster-41647.33265, Cluster-41647.4114), *hemH* (Cluster-41647.8307), and *HCCS* (Cluster-40119.0) were significantly lower in the yellow leaves than they were in the green leaves of *T. grandis* ‘Merrillii’, while the expression levels of the genes *hemE* (Cluster-41647.18465), *gltX* (Cluster-41647.32963), and *chlL* (Cluster-41647.30461) were higher in the yellow leaves than they were in the green leaves. The expression levels of the unigenes mentioned above are presented in Figure 4.

### 3.6. Validation of Transcription Data Using qRT-PCR

Differential expression analysis of RNA-seq was done using qRT-PCR technique, and six DEGs were used for the RT-PCR analysis. The expression tendency of these six genes by qRT-PCR was consistent with the data of RNA-Seq. (Figure 5), suggesting that the transcriptome data were highly reliable.

## 4. Discussion

### 4.1. Chlorophyll Deficiency Mutant

The etiolation phenomenon commonly occurs in plants, and has been systematically studied in crops and model plants. These mutants with yellow leaves are ideal material for further study on topics such as the development and function of chloroplasts and genetics and breeding [5,11]. Previous studies have reported that the composition and content of pigments in mesophyll cells are directly related to the formation of yellow leaf color in plants [12,21,22]. The contents of chlorophyll and carotenoid typically decrease simultaneously in the yellow leaves of woody plants [33,34]. In this study, the yellow leaf color variation of *T. grandis* ‘Merrillii’ was a chlorophyll-deficit mutant, and the carotenoid content did not differ between the green and yellow leaves. The yellow leaf mutant used in this study was obtained from a natural yellow leaf bud mutation, which is the first chlorophyll-deficit mutant identified in *T. grandis* ‘Merrillii’. The Chl *a/b* value was lower in the leaves of *T. grandis* ‘Merrillii’ with yellow leaves than it was with green leaves, which means that the deficiency of Chl *a* is more severe than that of Chl *b* and, thereby, inconsistent with previous studies. Usually the Chl *a/b* of wild-type leaves is higher than it is with mutant-type yellow leaves [22,35,36], leading to poor light harvesting ability and an underdeveloped light harvesting antenna system [37]. Our results found that the yellow leaf mutant of *T. grandis* ‘Merrillii’ was different from the etiolation mutant of other species, indicating that the structure and stability of the light-harvesting complexes and photosynthetic reaction centers or the size of the photosynthetic antenna may be changed [35,36,37,38]. This yellow leaf mutant is excellent material for studying chlorophyll synthesis, the structure of photosynthetic systems, and gene functions and their regulatory mechanisms in woody plants.

### 4.2. The Leaf Color Response of T. grandis ‘Merrillii’ Influences Chlorophyll Fluorescence

Photosynthetic electron transfer occurs in the thylakoid membranes of chloroplasts, and are related to PSI, PSII, and photosynthetic electron carriers [10,13,14,15]. Chlorophyll fluorescence is usually measured to investigate photosynthetic electron transfer [16,17,18]. This study measured the photosynthesis induction kinetics of PF, DF, and MR between green and yellow leaves of *T. grandis* ‘Merrillii’. In the yellow leaves of *T. grandis* ‘Merrillii’, PF kinetics changed significantly, and the signals (F_O_, F_J_, F_I_, and F_P_) decreased sharply. A previous study reported that decrease in the F_P_ signal may be related to an increase in non-radiative heat dissipation of the PSI antenna pigment, shrinking of the antenna pigment [39], a decrease in the number of active PSI reaction centers [40], a damaged PSI receptor side [18], or denatured and degraded photosynthetic pigment–protein complexes [38,39,40]. In the yellow leaves of *T. grandis* ‘Merrillii’, both the DF kinetic values decreased (Figure 2B,C), suggesting that the PSII electrons had a reduced ability to pass downstream and regenerate the antenna pigment [16].

The chlorophyll *a* fluorescence parameters in the leaves of mutant-type and wild-type *Torreya grandis* ‘Merrillii’ were used to illustrate photosynthetic energy and electron transfer. *F_v_/F_m_* represents the latent capacity of PSII quantum efficiency, and is widely used as an indicator of the photochemical activity of photosynthetic mechanism. *F_v_/F_m_* is considered to be an indicator of photoinhibition [41], and reflects the maximum quantum yield for primary photochemistry [42]. In this study, the ratio of Chl *a/b* in mutant-type *T. grandis* ‘Merrillii’ is lower than it is in wild-type plants (Table 1), which leads to stronger light-harvesting ability and a better light-harvesting antenna system. Meanwhile, the *F_v_/F_m_* value (Table 3) in the yellow leaves of mutant-type *T. grandis* ‘Merrillii’ decreased. ABS/RC values increased more in the leaves of mutant-type *T. grandis* ‘Merrillii’ than they did in wild-type leaves (Figure 3), suggesting that the antenna size became larger relative to P680 in the yellow leaves of *T. grandis* ‘Merrillii’ [43].

The φ_Ro_, φ_Eo_, ψ_0_, and γ_RC_ values decreased significantly in the mutant-type of *T. grandis* ‘Merrillii’ compared to the green leaves (Table 3, Figure 3), indicating a decrease in PQ exchange capacity at the Q_B_ site and in PQH_2_ reoxidation capacity [44], and a decrease in the quantum yield for electron transfer both from Q_A_^–^ to the electron transport chain beyond Q_A_^–^ and from Q_A_^–^ to the reduction of the end electron acceptors on the acceptor side of PSI in the yellow leaves of *T. grandis* ‘Merrillii’. The values of ET_0_/CS_M_ and RE_0_/CS_M_ were significantly lower in mutant-type *T. grandis* ‘Merrillii’ with yellow leaves compared to green leaves, and maybe both the size of the functional antenna and the exciton specific rate captured by the open RCS were reduced [45]. The decrease in ET_0_/CS_M_ and RE_0_/CS_M_ values and increase in DI_0_/CS_M_ are proposed to be due to the thermal inactivation of RCs [46]. These indicated that PSII activity limited linear electron transport in the yellow leaves of *T. grandis* ‘Merrillii’.

The changes in the redox states of P700 and PC were reflected by the changes in MR [47]. V_PSⅠ_, V_PSⅡ__-PSⅠ_ in mutant-type leaves changed faster than they did in wild-type plants (Table 2). This can be interpreted as the oxidation and reduction ability of PSI and PC, which increases when leaves mutate in color and become yellow [16]. Meanwhile, the value of δ_Ro_ has no significant difference in the yellow leaf mutant-type *T. grandis* ‘Merrillii’ compared to green leaves. These effects are inconsistent with the decrease in PSII activity in the leaves of *T. grandis* ‘Merrillii’, as analyzed above using the PF signal. These results illustrate that PSI activity was appreciably stimulated in the yellow leaves of *T. grandis* ‘Merrillii’. When PSII activity reduced, PSI activity increased, which could have been caused by the enhancement in cyclic electron flows [48].

Therefore, our results suggest that the potential quantum efficiency of PSII in yellow leaf mutant-type *T. grandis* ‘Merrillii’ decreased and damaged the photosynthetic apparatus, resulting in the inactivation of photosynthetic reaction centers and the degradation of D1 protein [41,42,49,50]. The photosynthetic electron carriers were blocked in the electron transfer from Q_A_^–^ to the electron transport chain beyond Q_A_^–^ and the electron transfer from Q_A_^–^ to the reduction of terminal electron acceptors on the acceptor side of PSI in the leaves of the yellow leaf mutant of *T. grandis* ‘Merrillii’. These results indicate that the cyclic electron flows around PSI may be a photo-protective mechanism of the yellow leaf mutant-type *T. grandis* ‘Merrillii’, which can help thylakoid membranes cope with the chlorophyll deficit and decreased linear electron transport of PSII.

### 4.3. The Leaf Color Response of T. grandis ‘Merrillii’ Influences Chlorophyll Synthesis

High-throughput transcriptome sequencing technology has been applied in several studies related to the molecular formation of yellow leaves in woody plants. One study reported that the molecular mechanism for the formation of the yellow leaf mutation in different woody species differed by identifying the expression level of the unigenes involved in chlorophyll metabolism, such as in *C. sinensis* ‘Baijiguan’ [21] and *L. indica* [33,34]. An example is the yellow leaf mutant of *L. indica*, which is caused by a disruption in coproporphyrinogen III oxidase (*CPO*) biosynthesis [34]. In the present study, 12 DEGs in the pairwise comparison between wild type (WT) and the yellow leaf mutant type (MT) were identified in ‘porphyrin and chlorophyll metabolism’ using the combined criteria of the log2 fold-change in the range between ≥ +2.0 and ≤ −2.0 and a corrected Padj value < 0.05. These 12 DEGs matched 10 genes (*hemC, hemE, hemF, hemH, hemY, ctaA, ctaB, gltX, HCCS,* and *UGT*), in which six unigene annotations in five genes (including *hemC, hemE, hemF, hemY, gltX*) were involved in chlorophyll synthesis (Figure 6A). In higher plants, the level of Glu TR mRNA (*gltX*) is positively correlated with chlorophyll synthesis [51]. Compared with green leaves, the expression of genes *hemC, hemE, hemF, hemY,* and *gltX* in yellow leaves of *T. grandis* ‘Merrillii’ were significantly downregulated, possibly because the chlorophyll synthesis precursor protoporphyrin IX had decreased, and then led to chlorophyll content decrease. Additionally, the unigenes involved in the synthesis of heme A and cytochrome were downregulated, as shown in Figure 6A. The effects of phytochrome and cytochrome on the formation of yellow leaves will be studied in future research.

Previous studies have shown that chlorophyll and carotenoid content is the proximate cause of yellow leaves [52]. This study determined that the chlorophyll content decreased, and the expression level of genes-related chlorophyll synthesis was downregulated in the yellow leaves of *T. grandis* ‘Merrillii’, while the carotenoid content and its synthesis displayed no significant changes compared to green leaves. Therefore, the leaf color mutation mechanism is mainly related to low chlorophyll synthesis efficiency and lower amounts of chlorophyll. Meanwhile, the PSII ability reduced, and the effective dissipation of excess excitation energy (DI_0_/CS_M_, DI_0_/RC, φ_Do_) increased, indicating that carotenoids play a protective role in the PSII of yellow leaves in *T.*
*grandis* ‘Merrillii’ from light damage [53].

### 4.4. The Leaf Color Response of T. grandis ‘Merrillii’ Influences Photosynthetic Electron Transfer

The photosynthesis that occurs in chloroplasts is usually separated into light reactions (light dependent) and dark reactions (light independent reactions). In photochemical reactions, PSII strips electrons from the water using light energy and releases O_2_. Electrons are transferred through the electron transport chain, ultimately producing NADPH. A previous study reported that chlorophyll fluorescence parameters (*F_v_/F_m_*) in the photochemical reactions of green leaves are significantly higher than those in yellow leaves [22], which was consistent with the results of this study. PF, DF, and MR have been used to investigate photosynthesis, especially the photochemical reaction in the leaves of plants [16,17,18]. This simultaneous measurement can collect and correlate complementary information for photosynthetic electron transport [19]. The present study found that the PF and DF were significantly lower, while MR was higher in yellow leaves than it was in the green leaves of *T. grandis* ‘Merrillii’ (Figure 2). Previous studies have shown that PF and DF predominantly emitted from antenna pigments of PSII, and MR, PF and DF kinetics depend on the redox state of the reaction center of PSI (P700) and PSII (P680) (Gao et al., 2014). Our study determined that the *F_v_/F_m_*, *ψ*_o_, γ_RC_, φ_Eo_, φ_Ro_, RC/CS_M_, ET_0_/CS_M_, and RE_0_/CS_M_ parameters in yellow leaves were significantly lower than in the green leaves of *T. grandis* ‘Merrillii’. These parameters reflect the changes in electron transfer rate on the PSII receptor side [53]. In summary, PSII has different electron transfer capacities in the yellow and green leaves of *T. grandis* ‘Merrillii’.

The expression of the *PsbR* gene (coding PSII 10-kDa protein) is blocked in the yellow leaves of *C. sinensis*, thus resulting in unstable PSII, abnormal chloroplast structure, and obstructed chlorophyll biosynthesis [21]. Thirty-seven unigenes associated with photosynthetic metabolism were identified based on the KEGG pathway in the yellow leaf mutant of *L. indica* [34]. However, only one DEG (ferredoxin-NADP^+^ reductase, *PetH*) was detected in the pairwise comparison between the yellow leaf samples (MT) and the control green leaf samples (WT) in the ‘photosynthesis’ metabolism when using the combined criteria |log2 fold-change| ≥ 1.0 and a corrected Padj value < 0.05 (Figure 6B). Expression of the *PetH* gene in the yellow leaves of *T. grandis* ‘Merrillii’ was significantly lower than it was in the green leaves. Interestingly, the expression of the *PetH* gene was upregulated in the yellow leaf color mutant of *L. indica* [34]. FNR is a photosynthetic electron carrier, catalyzing the terminal step of electron transport that is associated with PSI by transferring reducing equivalents from ferredoxin or flavodoxin to NADP ^+^ [54]. The expression of the *PsbA* in yellow leaves of *T. grandis* ‘Merrillii’ was significantly higher than that in green leaves. Our results considered that D1 protein degradation had occurred [39], which was consistent with the chlorophyll fluorescence parameter (*F_v_/F_m_*) in this study.

## 5. Conclusions

Compared to wild type, the chlorophyll content decreased and chlorophyll synthesis was blocked in the yellow leaves of *T. grandis* ‘Merrillii’. However, both carotenoid content and carotenoid synthesis demonstrated no significant changes between the green and yellow leaves of *T. grandis* ‘Merrillii’. Further, PSII activity was reduced, PSI activity increased, and the effective dissipation of excess excitation energy (DI_0_/CS_M_, DI_0_/RC, φ_Do_) increased in the yellow leaves of *T. grandis* ‘Merrillii’ compared to the green leaves. These results indicate that PSI activity and carotenoids play a photoprotective role in the yellow leaves of *T. grandis* ‘Merrillii’ to cope with decreased chlorophyll content and reduced PSII activity. The yellow leaf mutation in *T. grandis* is excellent material for *Torreya* breeding and research related to photosynthesis and chlorophyll. Furthermore, our study identified multiple DEGs, offering a global view of etiolation in *Torreya grandis*, which will help understand the formation process of yellow leaf and accelerate the molecular breeding of *Torreya grandis*.

## Figures and Tables

**Figure 1 cells-11-00431-f001:**
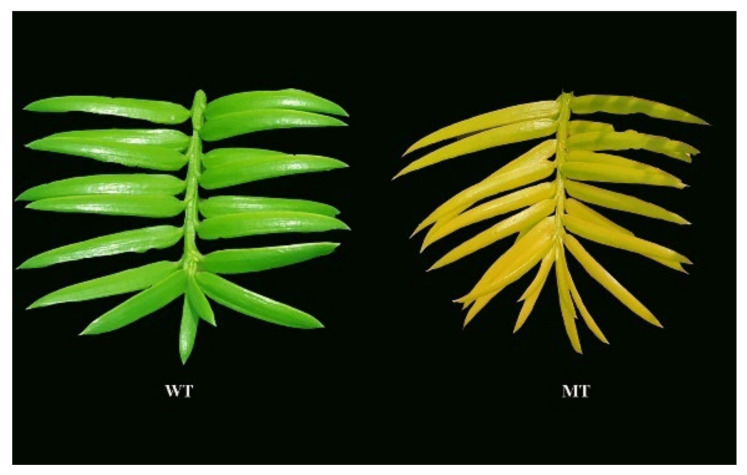
Branches of mutant-type and wild-type *Torreya grandis* ‘Merrillii’. (WT means wild type; MT means mutant-type yellow leaf).

**Figure 2 cells-11-00431-f002:**
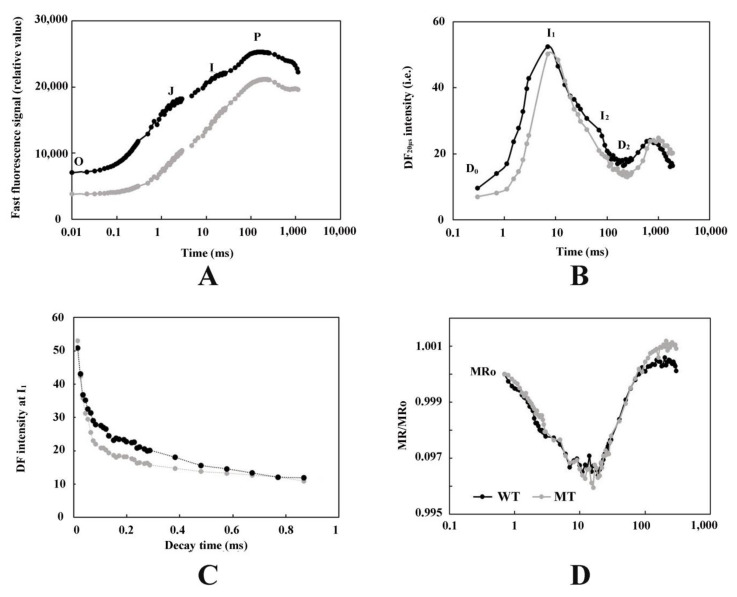
The PF, DF, and MR kinetics in leaves of MT and WT *Torreya grandis* ‘Merrillii’. Each curve is the average of three replicates. WT means wild type; MT means yellow leaf mutant type; (**A**), Prompt chlorophyll *a* fluorescence (PF). The signals were plotted on a logarithmic time scale. The signals used were fluorescence intensity at 20 µs ≡ F_o_; at 3 ms ≡ F_J_ and at 30 ms ≡ F_I_; maximum fluorescence intensity, F_P_ = F_m_. (**B**), Delayed chlorophyll *a* fluorescence. The signals were plotted on a logarithmic time scale. (DF). (**C**), The decay kinetics of DF at the characteristic maxima I_1_ (7 ms). The signals were plotted on a linear time scale. (**D**), Modulated 820 nm reflection (MR/MR_O_). The signals were plotted on a logarithmic time scale.

**Figure 3 cells-11-00431-f003:**
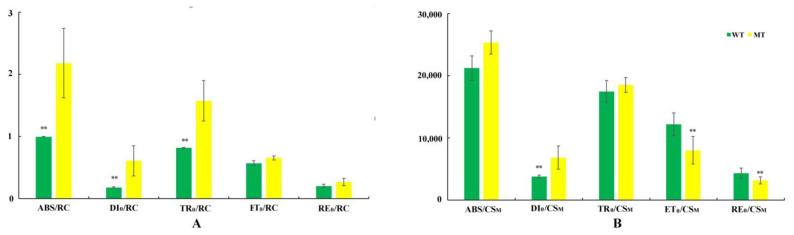
Energy flux models for the leaf samples of mutant-type and wild-type leaves under the same physiological conditions. The relative value of each parameter is proportional to the column height of the column charts. **Graph** (**A**) represents the specific activity on a single PSII reaction center (RC) basis. **Graph** (**B**) represents the specific activity on cross section (CS) basis. Bars represent the standard deviation of three repeats (*n* = 3); ** indicates a significant difference at *p* < 0.05.

**Figure 4 cells-11-00431-f004:**
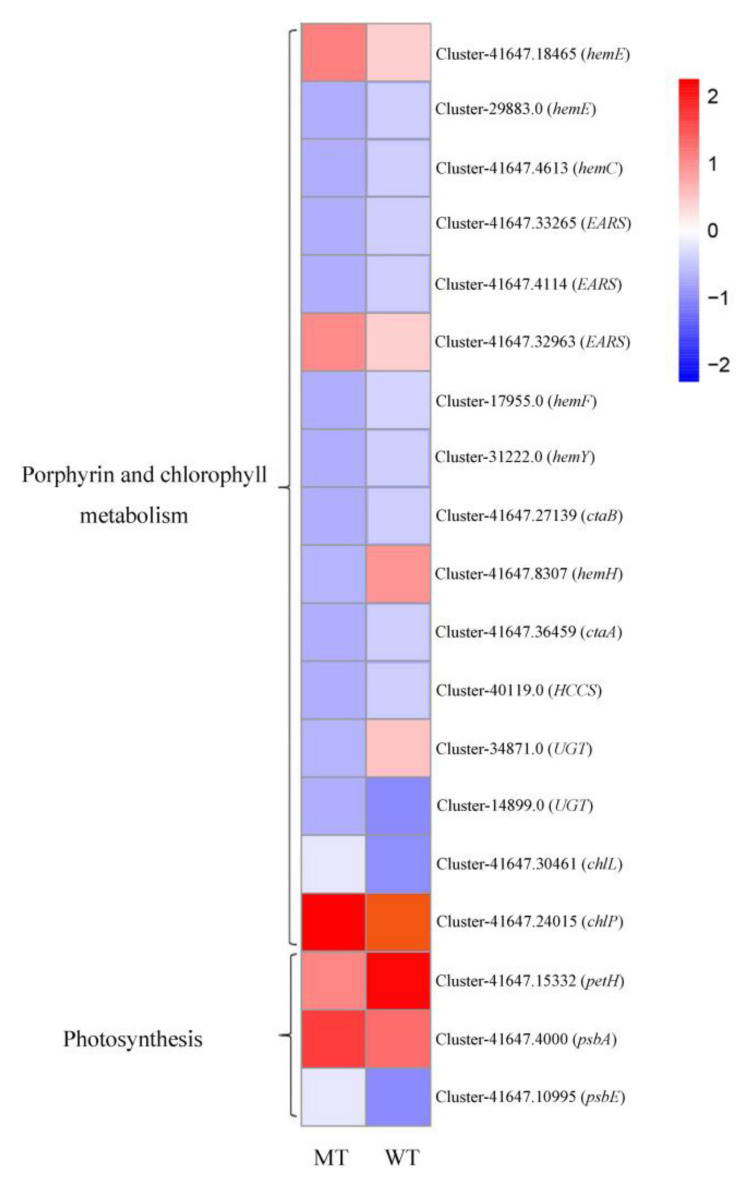
Heat maps of the differentially expressed unigenes involved in related pathways in leaves of mutant-type and wild-type *Torreya grandis* ‘Merrillii’. WT means wild type; MT means yellow leaf mutant type; the transcriptome data (FPKM) were used to create the heat maps.

**Figure 5 cells-11-00431-f005:**
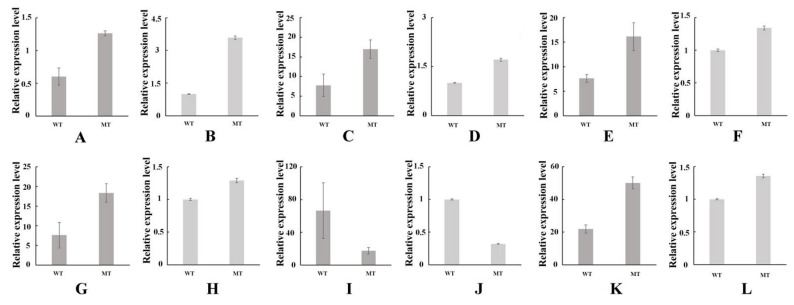
qRT-PCR confirmation of the DEGs identified by the transcriptome analysis. The relative expression level of each gene was expressed as the fold-change relative to the internal reference gene cyclophilin. WT means wild type; MT means yellowleaf mutant type; (**A**,**B**), unigene ID: Cluster-41647.30461. The relative expression of *chlL*. (**C**,**D**), unigene ID: Cluster-41647.18465. The relative expression of *hemE*. (**E**,**F**), unigene ID: Cluster-41647.32963. The relative expression of *EARS*. (**G**,**H**), unigene ID: Cluster-41647.29473. The relative expression of *VDE*. (**I**,**J**), unigene ID: Cluster-41647.15332. The relative expression of *petH*. (**K**,**L**), unigene ID: Cluster-41647.24000. The relative expression of *psbA*. (**A**,**C**,**E**,**G**,**I**,**K**), the data of RNA-Seq. (**B**,**D**,**F**,**H**,**J**,**L**), the data of qRT-PCR. The qPCR data of each sample with three biological and three technical replicates.

**Figure 6 cells-11-00431-f006:**
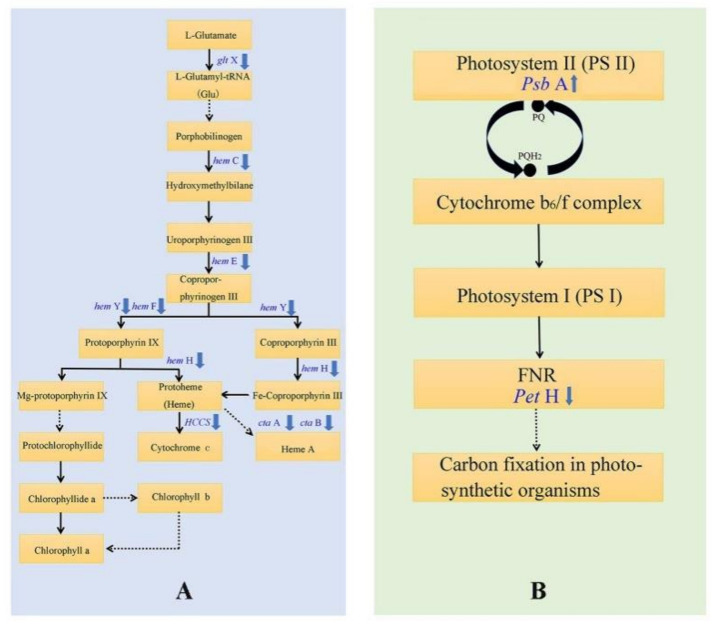
Genes involved in pathways related to the yellow leaf mutant of *Torreya grandis* ‘Merrillii’. (**A**), The porphyrin and chlorophyll metabolic pathway; (**B**), the photosynthetic pathway.

**Table 1 cells-11-00431-t001:** Photosynthetic pigment contents in leaves of mutant-type and wild-type *Torreya grandis* ‘Merrillii’.

*Torreya grandis* ‘Merrillii’	Chl *a* (mg/g)	Chl *b* (mg/g)	Chl (*a+b*) (mg/g)	Chl *a*/*b*	Carotenoid
WT ^1^	0.38 ± 0.02	0.09 ± 0.01	0.46 ± 0.03	4.36 ± 0.19	0.14 ± 0.01
MT ^2^	0.07 ± 0.00 ^** 3^	0.03 ± 0.00 ^**^	0.11 ± 0.00 ^**^	2.17 ± 0.33 ^**^	0.15 ± 0.01

^1^ WT means wild-type *T. grandis* ‘Merrillii’; ^2^ MT means yellow leaf mutant-type *T. grandis* ‘Merrillii’; ^3^ Each value is mean ± SE (*n* = 3), ^**^ indicates a significant difference at *p* < 0.05.

**Table 2 cells-11-00431-t002:** Parameters derived from the modulated 820 nm reflection (MR/MRo) of the mutant-type and wild-type *Torreya grandis* ‘Merrillii’.

*Torreya grandis*	V_PSⅠ_ ^3^ (0.7–20 ms)	V_PSⅡ-PSⅠ_ ^4^ (20–300 ms)	V_PSⅡ_ ^5^
WT ^1^	0.193 ± 0.01	0.015 ± 0.001	0.208 ± 0.014
MT ^2^	0.207 ± 0.003 ^**^	0.017 ± 0.001 ^**^	0.224 ± 0.004 ^** 6^

^1^ WT means wild-type *T. grandis* ‘Merrillii’; ^2^ MT means yellow leaf mutant-type *T. grandis* ‘Merrillii’; ^3^ V_PSⅠ_: maximum slope decrease of MR/MR_o_; ^4^ V_PSⅡ-PSⅠ_: maximum slope increase of MR/MR_o_; ^5^ V_PSⅡ_ = V_PSⅠ_ + V_PSⅡ-PSⅠ_; ^6^ Each value is mean ± SE (*n* = 3), ^**^ indicates a significant difference at *p* < 0.05.

**Table 3 cells-11-00431-t003:** Parameters derived from prompt chlorophyll *a* fluorescence transients in leaves of mutant-type and wild-type *Torreya grandis* ‘Merrillii’.

*Torreya grandis*	*F_v/_F_m_* ^3^	Ψo ^4^	δ_Ro_ ^5^	γ_RC_ ^6^	φ_Eo_ ^7^	φ_Ro_ ^8^	φ_Do_ ^9^	RC/CS_0_ ^10^	RC/CS_M_
WT ^1^	0.82 ± 0.01	0.70 ± 0.04	0.35 ± 0.02	0.50 ± 0.00	0.57 ± 0.04	0.20 ± 0.03	0.18 ± 0.01 ^**^	3804 ± 240	21,305 ± 2004
MT ^2^	0.73 ± 0.06 ^**^	0.43 ± 0.09 ^**^	0.41 ± 0.09	0.32 ± 0.06 ^**^	0.32 ± 0.09 ^**^	0.12 ± 0.02 ^**^	0.27 ± 0.06	3149 ± 254	12,146 ± 2875 ^** 11^

^1^ WT means wild type; ^2^ MT means yellow leaf mutant type; ^3^
*F_v_/F_m_*, maximal quantum efficiency of PSII; ^4^ V_J_, the relative variable fluorescence at the J-step; ψ_o_, a trapped exciton moves an electron into the electron transport chain beyond Q_A_^−^ at t = F_o_; ^5^ δ_Ro_, efficiency required for an electron to be transferred from reduced carriers between the two photosystems to the PSI end acceptors. ^6^ γ_RC_, a given chlorophyll *a* molecule that functions as the PSII reaction center. ^7^ φ_Eo_, quantum efficiency of electron transfer at t = F_o_. ^8^ φ_Ro_, quantum yield for reduction of the terminal electron acceptors on the PSI acceptor side. ^9^ φ_Do_, quantum yield for energy dissipation at t = F_o_. ^10^ RC/CS_o_ and RC/CS_m_, the Q_A_^−^reducing active RCs per CS (at t = F_o_; at t = F_m_). ^11^ Each value is the mean ± SE (*n* = 3), ^**^ indicates a significant difference at *p* < 0.05.

## Data Availability

Data Availability Statements in the “RNA-Seq Analysis” section. All reads used in this study were deposited in the NCBI Sequence Read Archive under accession number PRJNA687396.

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
