# Peer review of "Molecular and Photosynthetic Performance in the Yellow Leaf Mutant of Torreya grandis According to Transcriptome Sequencing, Chlorophyll a Fluorescence, and Modulated 820 nm Reflection"

_cells, 2022, doi:10.3390/cells11030431_

Round 1

Reviewer 1 Report

Molecular mechanism of photosynthetic energy and electron transfer in the yellow leaf mutant of Torreya grandis The manuscript is aimed to evaluate photosynthetic electron transport in yellow and green leaves of relict species Torreya grandis. The investigation is interesting, giving the new insight into physiological and molecular coordination of proteins involved in primary photosynthetic reaction in yellow Torreya mutant and it provides valuable information on synergic reaction between linear and cyclic electron transport especially in yellow conifer leaves. The importance of appearance of yellow leaves is really important to study since it involves different molecular mechanism of plastid development and chlorophyll biosynthesis. However, the authors fail to emphasize the importance and novelty of their study that should be included what is one of the major issue of this investigation. The authors should consider also also some other major issues to be addressed: The clearly stated hypothesis is missing and it should be included into the Introduction as well as the background that support the hypothesis. Abstract is the same as conclusion, both abstract and conclusion should be rewritten. Please find attached PDF file with comments and suggestions that I strongly suggest to be taken into consideration. Sincerely.

Author Response

Dear reviewers:

This is resubmission of the paper titled “Molecular mechanism of photosynthetic energy and electron transfer in the yellow leaf mutant of Torreya grandis ‘Merrillii’”.

We wish to thank you for the time and effort you have spent reviewing our paper. We are pleased to note that you have found our research work interesting and also pointed out some problems to help us improve the quality of our work. 

Motivated by your comments, we have deeply tried to fix all the problems you mentioned in manuscript.

Sincerely,

Songheng Jin,

Jiyang College, Zhejiang A&F University, Zhuji, Zhejiang, 311800, China

[email protected]

Reviewer 2 Report

The MS focused on photosynthetic acclamatory responses in yellow leaf mutants of T. grandis. The authors sufficiently review literature and justified the objectives of the study.  The authors well planned the experiments in view of their objectives and measured appropriate attributes such as PSII stability and functionality test, PSI activity tests, photosynthetic pigment tests, and transcriptome analysis. The most important complementary dataset from transcriptome analysis suggested that mutant of T. grandis adjusted the biosynthesis of photosynthetic pigments to regulate PSII and PSI activity. Ratio of carotenoids to chlorophyll was substantially increased from 1:4 to 1:1 supported this argument. In view of transcriptome analysis  for biosynthetic pathways of carotenoids and chlorophyll, increase in amount of carotenoids and down regulation of PSII activity along with increase in heat dissipation per reaction center, it is suggested that carotenoids play a significant role in photoprotection of PSII. However, being a reader, I missed the Discussion in this context. Instead the authors focused on role of cyclic electron transport dependent development of pH gradient and photo-protection. However, the authors did not presented sufficient evidences to conclude this statement "Cyclic electron flows around PSI as an photo-protective mechanism in yellow leaves mutant type of T. grandis, can help the thylakoid membranes maintained the gradient (ΔpH) coping with the chlorophyll-deficit and the linear electron transport decreased of PSII." If the authors insist on their conclusion, it is suggested to present the data for PSI quantum yield, electron transport through PSII & PSI, and calculate the cyclic electron transport --- and finally draw the relationship between PSII activity and Cyclic Electron Transport. 

Overall, the MS is nicely written and can be accepted after minor revision in Discussion section as suggested earlier.

Author Response

Dear reviewers:

This is resubmission of the paper titled “Molecular mechanism of photosynthetic energy and electron transfer in the yellow leaf mutant of Torreya grandis ‘Merrillii’”.

We wish to thank you for the time and effort you have spent reviewing our paper. We are pleased to note that you have found our research work interesting and also pointed out some problems to help us improve the quality of our work. 

Motivated by your comments, we have deeply tried to fix all the problems you mentioned in manuscript.

Sincerely,

Songheng Jin,

Jiyang College, Zhejiang A&F University, Zhuji, Zhejiang, 311800, China

Reviewer 3 Report

Line 9 and following: Abstract should, by itself, be comprehensible. Therefore symbols which are not very commonly used and well-known, should not be used without explanation.

Line 11: transcriptomic > transcriptomics?

Line 39–40: Linebreak without reason

Line 60: mutation > mutant     (> means “change to)

Line 78: What is meant with “respectively”?

Line 101: 820 ± 25 nm; line 103: 820 ± 20 nm);  are the values after ± “halfband widths”?

Line 109–110: “after actinic illumination”; do you mean “after start of actinic illumination”?

Line ≈162.3, Table 1: Chl a, Chl b; use of italics unusual; usually people write Chl a, Chl b.

It is also inconsistent to have a and b as italics only in the fifth column. Dividing 0.38 by 0.09 makes 4.22, not 4.36; please check! 0.07 divided by 0.03 makes 2.33, not 2.17; please check!

0.07 plus 0.03 makes 0.10, not 0.11; please check.

Line 162. I would write present tense, i.e. “show” instead of “showed” in places like this.

Line 167, 170, 172, 174 “Figure 2a” etc.: Use capitals “Figure 2A” etc., since this is how you have labelled the figures.

Lines 174–175:” The lowest points of the 174 MR/MRO kinetics of mutant type appeared earlier than wild type.” I cannot see this in the figure; also grammar needs correction.

Line 177: “Torreya grandis” italicize “Torreya grandis”!

Line 179: “20μs”, “300μs”; make space between digits and unit: “20 μs”, “300 μs”!

Figure 2: It would be nice to have some kind of (a few) “error bars” shown in the Figure, since 3 is a very small number of replicates.

Line 187: “than wild type” > “than that of the wild type”.

Line 217: “were” > “are”

Line 220, 223: “T. grandis” > “T. grandis”.

Line 246: “(Qphred <= 20)”. Not clear without explanation.

Line 360: “chlorophyll a” should be “chlorophyll a”.

Author Response

(The authors gave the same response as above.)

Round 2

Reviewer 1 Report

Thank you for taken into consideration most of my comments, however, there are few more corrections that needs to be done. The hypothesis is still missing, the figures quality is still quite poor, some subtitles in the Discussion section needs to be changed and the conclusion needs to be rewritten. Please, find the attached PDF file with comments that needs to be corrected.

Best regards 

Author Response

Dear reviewers:

This is resubmission of the paper titled “Molecular mechanism of photosynthetic energy and electron transfer in the yellow leaf mutant of Torreya grandis”.

We wish to thank you for the time and effort you have spent reviewing our paper. We have fix all the problems you mentioned in manuscript.

Please, find the attached Word file with our response.

Songheng Jin 

Jiyang College, Zhejiang A&F University, Zhuji, Zhejiang, 311800, China
